# Surgical Management of Jugular Foramen Schwannomas

**DOI:** 10.3390/cancers13164218

**Published:** 2021-08-22

**Authors:** Amir Kaywan Aftahy, Maximilian Groll, Melanie Barz, Denise Bernhardt, Stephanie E. Combs, Bernhard Meyer, Chiara Negwer, Jens Gempt

**Affiliations:** 1Department of Neurosurgery, School of Medicine, Klinikum rechts der Isar, Technical University Munich, 80333 Munich, Germany; maxi.groll@gmx.de (M.G.); melanie.barz@tum.de (M.B.); bernhard.meyer@tum.de (B.M.); Chiara.Negwer@tum.de (C.N.); jens.gempt@tum.de (J.G.); 2Department of Radiation Oncology, School of Medicine, Klinikum rechts der Isar, Technical University Munich, 80333 Munich, Germany; Denise.Bernhardt@mri.tum.de (D.B.); stephanie.combs@tum.de (S.E.C.); 3Department of Radiation Sciences (DRS), Institute of Radiation Medicine (IRM), Helmholtz Zentrum München (HMGU), Ingolstädter Landstraße Ingolstädter Landstraße 1, 85764 Oberschleißheim, Germany; 4Deutsches Konsortium für Translationale Krebsforschung (DKTK), Partner Sites Munich, 80333 Munich, Germany

**Keywords:** jugular foramen tumor, skull base surgery, surgical technique, approach, neuro-oncology, schwannoma

## Abstract

**Simple Summary:**

Treatments of jugular foramen schwannomas may be challenging due to critical anatomical relations and the involvement of different aspects of the skull base. Advances in microsurgery have led to improved outcomes over recent decades, whereas in contrast, some advocate stereotactic radiotherapy as an effective therapy, controlling the tumor volume with few complications. In this manuscript, we present the outcomes and adverse events in a contemporary cohort and discuss surgical advantages and disadvantages of different performed classic skull base approaches.

**Abstract:**

Background: Resection of jugular foramen schwannomas (JFSs) with minimal cranial nerve (CN) injury remains difficult. Reoperations in this vital region are associated with severe CN deficits. Methods: We performed a retrospective analysis at a tertiary neurosurgical center of patients who underwent surgery for JFSs between June 2007 and May 2020. We included nine patients (median age 60 years, 77.8% female, 22.2% male). Preoperative symptoms included hearing loss (66.6%), headache (44.4%), hoarseness (33.3%), dysphagia (44.4%), hypoglossal nerve palsy (22.2%), facial nerve palsy (33.3%), extinguished gag reflex (22.2%), and cerebellar dysfunction (44.4%). We observed Type A, B, C, and D tumors in 3, 1, 1, and 4 patients, respectively. A total of 77.8% (7/9) underwent a retrosigmoid approach, and 33.3% (3/9) underwent an extreme lateral infrajugular transcondylar (ELITE) approach. Gross total resection (GTR) was achieved in all cases. The rate of shunt-dependent hydrocephalus was 22.2% (2/9). No further complications requiring surgical intervention occurred during follow-up. The median follow-up time was 16.5 months (range 3–84 months). Conclusions: Considering the satisfying outcome, the GTR of JFSs is feasible in performing well-known skull base approaches. Additional invasive and complicated approaches were not needed. Radiosurgery may be an effective alternative for selected patients.

## 1. Introduction

Jugular foramen schwannomas (JFSs) are rare, originating from cranial nerves (CNs) IX, X, and XI. They constitute approximately 2.9–4% of all intracranial schwannomas [1,2]. Surgery involving this area usually puts the lower CNs at risk. Reoperation in this vital region is associated with severe CN deficits [3]. Thus, the primary goal of surgical JFS management is to achieve total resection while preserving CN function. Only a few surgical series have included more than 20 cases of JFSs due to their rarity [3,4,5,6,7,8,9,10].

Surgical management of JFSs is also defined by their anatomical characteristics. Tumors originating proximally tend to have a more intracranial growth pattern; those originating in the mid-region expand into the bone; and distal-originating lesions often present with extracranial growth. These various patterns have been used to classify them into subtypes. Although several classification systems have been proposed [1,4,11], Kaye’s [11], Pellet’s [12], and Samii’s [2] classifications are the most commonly used (Table 1). Type A tumors are the most common types, presenting with vestibulocochlear dysfunction. Type C tumors are the least common. They present with ninth and twelfth CN deficits, mainly hoarseness and decreased gag reflex [13].

Recently, typical surgical management strategies have become more conservative due to the increasing use of radiation as the primary treatment modality [14]. However, the most effective method for treating JFSs remains to be determined. In this manuscript, we present the outcome and adverse events in a cohort of JFS and discuss the surgical advantages and disadvantages of fundamental skull base techniques.

## 2. Materials and Methods

### 2.1. Study Design and Outcome Parameters

We performed an observational retrospective single-center case series study. Patients who underwent surgery for JFSs between June 2007 and May 2020 were included. We analyzed the clinical records of patients according to the surgical approach, pre- and postoperative neurological status, Karnofsky Performance Status Scale (KPSS), and adverse events during follow-up visits. We divided JFSs according to Samii’s classification [2]. Then, we determined the extent of resection by means of pre- and postoperative T1 ± contrast agent 3.0 T MRI sequences.

### 2.2. Statistics

We performed statistical analysis using the software STATA (version 13.1, 2011, StataCorp LLC, 4905 Lakeway Drive, College Station, Texas 77845-4512, TX, USA). Normal distribution was assumed according to the central limit theorem. Data in text and graphs are shown as the median (mdn.) with interquartile range (IQR) or mean ± standard deviation (SD).

### 2.3. Surgical Approaches

#### 2.3.1. Retrosigmoid Approach

Analogous to the pterional approach for the anterior skull base, the retrosigmoid approach is the workhorse approach regarding the posterior fossa and the cerebellopontine angle, already described in the literature in detail [15,16,17,18], whereas we prefer a C-shaped skin incision (Figure 1).

#### 2.3.2. Extreme Lateral Infrajugular Transcondylar (ELITE) Approach

The ELITE approach involves resection of the medial and superior medial part of the occipital condyle and jugular tubercle. The anterolateral modification includes the addition of a high cervical exposure. Other modifications include a limited transcondylar approach, partial resection of the occipital condyle, and resection of the C1 arch for access to chordomas, chondrosarcomas, and high cervical spine lesions. Such modifications would go beyond the scope of this manuscript.

Patient positioning and incision differ for the dorsolateral and anterolateral technique (Figure 2). For the dorsolateral ELITE approach, the patient is placed in the same position as for a classic retrosigmoid method. For the anterolateral technique, the patient is in a supine position with the head displaced away from the side of the lesion.

## 3. Results

### 3.1. Patient Population

Nine patients underwent resection for JFS and were analyzed. The median age was 43 years (range 20–71 years). The initial symptoms for which they sought medical help included hearing loss (66.6%), headaches (44.4%), hoarseness (33.3%), and dysphagia (44.4%). In all cases of hoarseness, unilateral vocal cord paresis was documented by fiberoptic laryngoscopy, also to assess perioperative possible morbidity and to clearly discuss postoperative worsening with the patient. On examination, we observed involvement of the hypoglossal nerve with weakness of the tongue in 22.2% of patients, facial nerve palsy in 33.3% of patients, extinguished gag reflex in two patients (22.2%), and cerebellar dysfunction in 44.4% of patients (Table 2).

### 3.2. Postoperative Outcome

The median follow-up time was 16.5 months (range 3–84 months). Surgery-related mortality was 0% (Table 3). Gross total resection (GTR) was achieved in all cases, whereas two patients had already undergone an operation and received radiotherapy. Two type D tumors and one type A tumor had a cystic configuration; none of them underwent previous treatment. Postoperatively, new permanent hoarseness (with intact gag reflexes), hearing loss, and mild facial nerve palsy (House and Brackmann grade II) occurred in one patient, each. Thus, 33.3% of the patients suffered from new permanent deficits, whereas the others recovered well from transient hoarseness in 22.2% (2/9), facial nerve palsy in 11.1% (1/9), and vertigo and gait disturbance in 44.4% (4/9) during follow-up. Two patients with cystic JFS suffered from above-mentioned hoarseness and the facial nerve palsy, although intraoperatively, no complications occurred, and both lower CN and CN VII monitoring were uneventful. Cerebellar dysfunctions, dizziness, and headaches improved in all cases, and hoarseness improved in 50.0% of patients after surgery. One patient with a type D tumor suffered from postoperative lung artery embolism but adequately recovered during the hospital stay. The rate of shunt-dependent hydrocephalus was 22.2% (2/9). No further complications requiring surgical intervention occurred during follow-up.

## 4. Discussion

JFSs are rarely seen. A greater awareness of the natural history of intracranial schwannomas, such as vestibular or trigeminal schwannomas, has developed, leading to increased consideration of radiosurgery as an alternative modality for treating these tumors [14,19].

The purpose of the present study was to analyze the outcomes and comorbidities associated with radical surgical resection with feasible techniques.

### 4.1. Choosing a Suitable Approach

The treatment strategy for JFSs should always respect the patient’s anatomy, clinical presentation, and baseline characteristics. The size of the tumor, goal of the surgery (e.g., biopsy, brainstem decompression, GTR), and characteristics of the tumor must be considered. Our results suggest that single-stage GTR of JFSs is preferable and that it can be achieved without severe complications in the majority of patients.

Various modified surgical approaches have been reported as better treatment options for these tumors [2,4,7,20,21,22]. In general, approaches to the jugular foramen can be divided into three groups: lateral, posterior, and anterior [3,23]. Most JFSs are resected through the lateral or posterior trajectory. Al-Mefty et al. and Samii et al. argued that hearing improvements can be observed following JFS surgery [24,25]; therefore, techniques causing hearing loss may be avoided (e.g., a translabyrinthine approach). The preferred surgical approaches for JFSs have progressed to the more precise removal of the affected structures, as we show in our series.

The main techniques in exposing the jugular bulb involve infralabyrinthine mastoidectomy and resection of the jugular process [22,26]. Some prefer the petro-occipital trans-sigmoid approach [3,6], suboccipital transjugular process approach, or paracondylar–lateral cervical approach for JFSs [3]. In contrast, we preferred to use a classical retrosigmoid approach as much as possible [1,2,9,15,25,27]. With this conventional technique, it is not necessary to ligate the sigmoid sinus, and the cerebellopontine cistern and internal gate of the jugular foramen can be exposed in a satisfying fashion. When JFSs extend into the foramen magnum, the ELITE approach can be used [28]. It can be seen as a continuous development of the retrosigmoid and, in particular, the transcondylar approach, which was first proposed as an access point to the foramen magnum and ventral medulla [22,28]. The approach can be individually tailored to provide the necessary exposition. The literature has adequately described two modifications of this approach that have been performed at our institute: the dorsolateral ELITE (Figure 3 and Figure 4) used for JFS with a large intradural component (Type A, B) and the anterolateral ELITE approach (Figure 5) used for Type C and D tumors [7,22]. Several authors have demonstrated that the technique [3,7,10,23,29] is not associated with obvious approach-related complications (e.g., facial palsy or vertebral artery injury) when compared with infralabyrinthine approaches.

However, infralabyrinthine mastoidectomy is still widely used [2,3,12,25]. It provides access between the labyrinthine area and the dome of the jugular bulb, laterally exposing the jugular foramen, whereas the hearing function is still at risk. Endoscopic techniques have been proposed that require drilling of the suprajugular bone using a technique similar to the access to the internal auditory canal in patients with vestibular schwannomas [24,27]. In our series, we mainly counted on the classic retrosigmoid technique and its extension, the ELITE approach, with satisfactory results.

We experienced some occult difficulties with two cases with cystic schwannomas. Two such cases suffered from hoarseness and facial nerve palsy, although intraoperatively, no complications occurred, and CN monitoring was uneventful. Carvalho et al. described that schwannomas usually do not infiltrate the CNs, permitting radical resection; however, for cystic schwannomas with a thin cyst wall, the arachnoid plane may not be preserved after opening the cyst which makes resection much more difficult. We do agree with these obstacles; one tends to more aggressive techniques. Therefore, surgeons should begin identifying the interface between tumor and its surrounding structures and avoid opening the cyst first. Respecting the cystic nature and using such techniques, postoperative morbidity of cystic schwannomas could be seen less frequently than solid tumors [29,30].

Nevertheless, one optimal surgical approach cannot be pinpointed, because this is also determined by experience and preference. Intraoperative neuro-electrophysiological monitoring is of utmost importance regarding preserving neurological function and predicting postoperative neurological outcomes.

### 4.2. Extent of Resection and Functional Outcome

Due to a lack of references and concerns regarding CN injury, the optimal treatment for JFSs remains controversial. Most authors have supported the fact that microsurgical GTR is the first-line therapy [3,5,26]. Others have claimed that single-stage techniques such as reoperations may cause severe CN deficits due to scarring from earlier surgeries [2,5,25]. Some have even suggested leaving the tumor adjacent to CNs or the brainstem to prevent severe neurological deficits [5]. In contrast, Sedney et al. demonstrated that subtotal resection increases surgical morbidity without a significant increase in tumor recurrence [7].

Advancements in microsurgical techniques have improved the surgical outcome. Nevertheless, microsurgery for JFSs still carries a relatively high risk of surgery-related morbidity and lower CN deficits. When reviewing the literature, the rates of postoperative CN morbidity do vary. This is likely explained by the selected surgical approach. Invasive and complicated procedures, such as transposing the facial nerve or sacrificing the labyrinth and cochlea, provide a wide exposure but also lead to additional CN morbidity [3,25,26]. Worsening or newly developed CN deficits are the most common postoperative morbidity, as already seen in previous studies [3,24]. Of our patients, 33.3% suffered from new postoperative CN deficits.

As an alternative, stereotactic radiosurgery was shown to improve risk profiles in patients with residual or newly diagnosed small-volume JFSs [6]. Multicenter studies have suggested that gamma knife surgery can achieve excellent tumor control and improvements in neurological function in most patients with either primary or residual JFSs [1,31,32]. However, with only a small number of cases and a comparatively short follow-up period of treating JFSs with radiosurgery, the evidence remains limited [33]. Stereotactic radiosurgery and fractionated radiotherapy have been performed for patients with intracranial schwannomas from different origins. Larger tumors with brainstem compression should primarily be treated with surgery rather than radiosurgery [34]. However, if surgery is contraindicated, excellent tumor control rates at 5 and 10 years were achieved in 94% of patients with trigeminal schwannoma [31].

For small vestibular schwannomas, e.g., a high rate of tumor growth control (>95%) of 5 or more years after radiotherapy with lower morbidity compared to surgery can be expected [32,35]. Consequently, fractionated radiotherapy and stereotactic radiosurgery as an alternative to surgical resection for patients with smaller intracranial schwannomas, both vestibular and non-vestibular, are alternative treatment options. Tumor control rates for trigeminal schwannoma and jugular foramen tumors are excellent, with actuarial tumor control rates of >95% at 5 and 10 years in both stereotactic radiosurgery and fractionated radiotherapy, with minimal morbidity and toxicity [31,36,37,38,39]. The use of radiotherapy to treat trigeminal schwannoma resulted in functional improvement in 67.3% of patients, stable lesions were found in 26.9% of patients, and worsening of the disease occurred in only two patients (3.8%) [40]. Retrospective analyses of patients treated with radiosurgery for jugular foramen tumors showed lower cranial injury after radiosurgery compared to surgery, and radiosurgery was the preferred management strategy if patients did not have large tumors and symptomatic mass effect. [41]. Interestingly, foramen jugular tumors showed a substantial decrease in tumor volume in about 50% of cases [39]. Additionally, overall preservation or improvement in cranial nerve function was noted in 98% of motor cranial nerves.

However, there appears to be an increased risk of transient enlargement and increased toxicity of large, cystic lesions undergoing radiotherapy [31]. The median tumor volume was 7.08 cm^3^ and median maximal diameter was 3.10 cm in our series. We observed that all patients were symptomatic at time of presentation and had relatively large space-occupying tumors. According to interdisciplinary tumor board discussions, primary surgical treatment was indicated first. We did not observe any incidental findings of a JFS; all patients experienced a decrease in their quality of life, with cranial nerve deficits as well. Regarding tumor size, a systematic review aimed to compare outcomes of surgery and stereotactic radiotherapy with no earlier intervention; pooled analysis demonstrated that stereotactic radiotherapy is superior to surgery in cases of vestibular schwannomas with diameters of less than 3 cm. Both approaches were comparable in terms of tumor control. A cohort study compared the outcomes of stereotactic radiotherapy and surgery in vestibular schwannomas with a tumor diameter ≤ 2.8 cm with no differences observed in terms of functional outcomes, tumor control and mortality [42,43,44]. Mathieu et al. have treated 62 of NF2 patients using radiosurgery. The mean tumor volume here was 5.7 cm^3^, and serviceable hearing was present in 35%. The control and the hearing preservation rates were observed to be 85%, 81% and 81% at 5, 10 and 15 years, and 73%, 59% and 48% at 1, 2 and 5 years, respectively. Tumor volume was significantly predictive of local control [45]. In another analysis, Phi et al. reported about 36 NF2 patients treated with radiosurgery with a mean tumor volume of 3.2 cm^3^. Five patients developed tumor recurrence, and the calculated control rates were 81%, 74% and 66% in the first, second, and fifth year, respectively [44,46].

None of our patients received postoperative radiotherapy, the extent of resection was sufficient, and the functional outcome was satisfying. Surgery with the aim of GTR of JFSs is a valid treatment option, especially in larger lesions due to the abovementioned facts Patients with JFS tend to present at a very late state, where cranial nerve compression or other tumor mass-induced deficits have already occurred. With reference to tumor size and volume, patients with JFS tend not to be the perfect cases for primary radiotherapy instead of primary surgical resection and debulking. However, in the case of smaller lesions, radiotherapy has to be discussed as a possible alternative.

Regarding a “wait and watch strategy”, all our decisions are made according to interdisciplinary tumor board discussions. Of course, in the case of complete asymptomatic patients and small tumors, imaging controls may be discussed together with the patients, whereas as mentioned above, in such cases, which we did not experience at all, a primary radiotherapy could be another rational modality instead.

Bakar et al. reported some technical related complications such as CSF leakage (6.5%), meningitis (2.0%), aspiration pneumonia (1.5%) and mastoiditis (1%) [30]. Vertebral artery injury was developed in two patients (1.0%), hemiparesis was seen in two patients (1.0%), and one patient (0.5%) died after a complicated CSF leakage followed by meningitis. Such rates could also occur in our institute; we just experienced a few cases of JFS. Maybe we also did not experience such complications because we focused on a few but well-experienced techniques. This also one of the main aims of the manuscript; a proposal to other surgeons to concentrate on a few feasible techniques in order to reduce postoperative morbidity and mortality. According to the review, complete tumor removal was achieved in 159 patients (86.9%), near-total tumor resection was achieved in 6 patients (3.3%), and subtotal tumor removal was accomplished in 18 patients (9.8%). We achieved complete removal in all cases (100%). Of course, tumor size and tumor characteristics do play an important role, too.

Nowadays, with superior preoperative visualization due to high-resolution imaging, a valid choice of approach can be made during presurgical planning. Most of the jugular foramen region can be reached and overseen with fewer, yet standard approaches, avoiding risk to the crucial structures. There is not a superior or “one fits all” approach to JFSs; however, based on our series, we think the “few fit most” concept of using classic approaches promotes a very good outcome with a high learning curve in the majority of cases.

### 4.3. Study Limitations

This was a retrospective case series; therefore, it was not possible to determine causalities with respect to clinical outcome. We acknowledge several limitations, including the small number of patients and relatively short follow-up time. To establish a solid conclusion on the optimal treatment strategy for JFSs, further validation with larger cohorts and longer follow-up times are needed.

## 5. Conclusions

Considering the low operative morbidity and satisfying functional outcome, GTR of JFSs is feasible through performing well-known approaches. Additional, more invasive, and complicated techniques were not needed in our series. Radiosurgery is an effective alternative for satisfactory functional outcome with adequate oncologic control in patients with small lesions and who are not suitable for surgery. Early diagnosis, adopting the appropriate surgical approach, and GTR of these benign tumors, can result in satisfactory surgical outcomes.

## Figures and Tables

**Figure 1 cancers-13-04218-f001:**
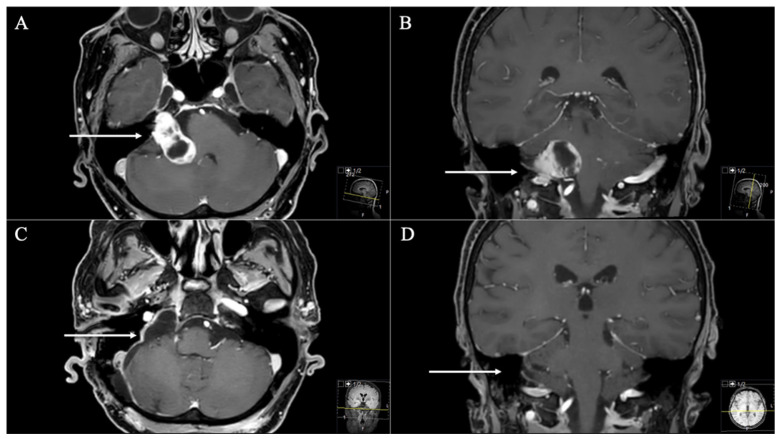
A 66-year-old female patient presenting with headache, vertigo, and cerebellar dysfunction. (**A**) Preoperative axial and (**B**) coronal T1-weighted gadolinium-enhanced MRI, displaying cystic JFS (arrows), involving the CPA (Samii Type A). (**C**) Postoperative axial and (**D**) coronal MRI control, indicating complete resection (arrows) through a classic retrosigmoid approach. Postoperatively, mild facial nerve palsy (House and Brackmann II) occurred and remained during follow-up.

**Figure 2 cancers-13-04218-f002:**
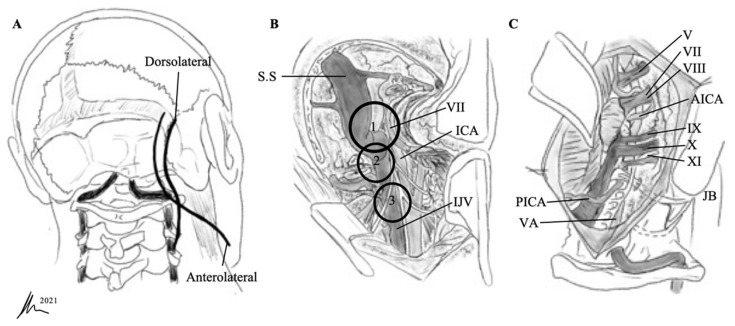
(**A**) For the dorsolateral approach, a lazy S or bigger C incision is used, about 1 cm posterior to the mastoid bone, extending inferiorly along the hairline. For the anterolateral ELITE procedure, a retroauricular curvilinear C-shaped, or question-mark-shaped, skin incision is begun approximately 2 to 3 cm posterior to the upper border of the ear. Inferiorly, this incision is carried down into the neck, traversing the border of the sternocleidomastoid muscle and running parallel to the body of the mandible (fingerbreadths below). For the dorsolateral approach, the sternocleidomastoid muscle is retracted anteriorly; for the anterolateral approach, it is retracted posteriorly. (**B**) Exposure by different techniques: (1) suprajugular, infralabyrinthine; (2) infrajugular, transcondylar; and (3) high cervical. SS, sigmoid sinus; ICA, internal carotid artery; IJV, internal jugular vein. (**C**) Anterolateral techniques with high cervical dissection. AICA, anterior inferior cerebellar artery; PICA, posterior inferior cerebellar artery; VA, vertebral artery; JB, jugular bulb.

**Figure 3 cancers-13-04218-f003:**
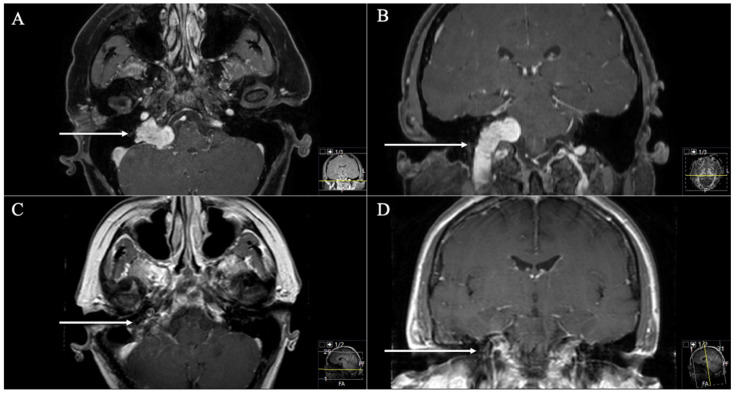
A 56-year-old female patient presented with headache, cerebellar dysfunctions, hearing loss, hoarseness, and extinguished gag reflex. (**A**) Preoperative axial and (**B**) coronal T1-weighted gadolinium-enhanced MRI, showing an impressive JFS (arrows) with a quasi-dumbbell shape and both intra- and extracranial components through the JF (Samii Type D). (**C**) Postoperative axial and (**D**) coronal MRI control, indicating complete resection (arrows) through a dorsolateral ELITE approach. Postoperatively, no new deficits occurred, and the patient recovered from the cerebellar dysfunctions, hoarseness, and extinguished gag reflex.

**Figure 4 cancers-13-04218-f004:**
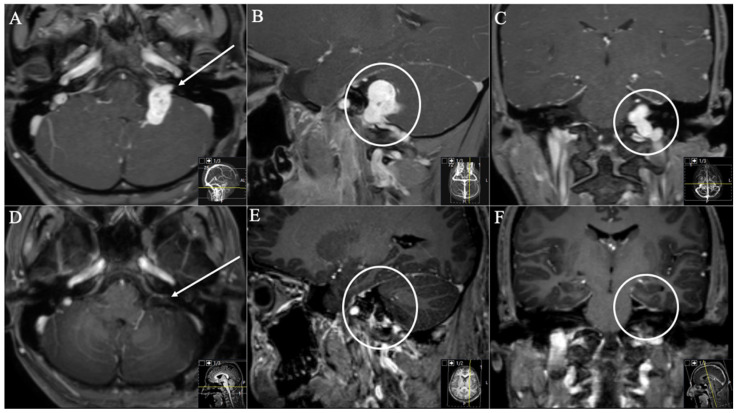
A 26-year-old male patient presented with progressive hoarseness, dysphagia, and extinguished gag reflex. (**A**) Preoperative axial (arrow), (**B**) sagittal (circle), and (**C**) coronal (circle) T1-weighted gadolinium-enhanced MRI, showing the JFS with intracranial extension (Samii Type B). (**D**) Postoperative axial, (**E**) sagittal, and (**F**) coronal MRI control, indicating complete resection through a dorsolateral ELITE approach. Postoperatively, no new deficits occurred. He suffered from postoperative temporary vertigo; hoarseness showed slight improvement during follow-up.

**Figure 5 cancers-13-04218-f005:**
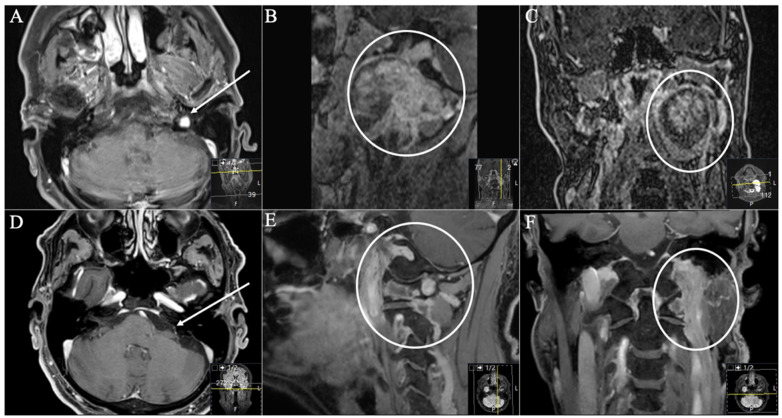
A 46-year-old male patient presented with headache, hoarseness, dysphagia, and extinguished gag reflex. (**A**) Preoperative axial (arrow), (**B**) sagittal (circle), and (**C**) coronal (circle) T1-weighted gadolinium-enhanced MRI, showing a massive space-occupying extracranial jugular foramen schwannoma with extension into the JF (Samii Type C). (**D**) Postoperative axial, (**E**) sagittal, and (**F**) coronal MRI control, indicating complete resection through an anterolateral ELITE approach. Postoperatively, no new deficits occurred, and the patient recovered from the hoarseness and extinguished gag reflex. Dysphagia remained but improved during follow-up as well.

**Table 1 cancers-13-04218-t001:** Classifications for JFS. Data are shown as CPA, cerebellopontine angle; JF, jugular foramen; JFS, jugular foramen schwannoma.

	Samii et al. [2]	Kaye et al. [11] and Pellet et al. [12]
A	JFS in the CPA with minimum enlargement of JF	JFS in the CPA with minimum enlargement of JF and with a small extension into the bone
B	Main portion at JF with intracranial extension	JFS invading the bone with or without intradural parts
C	Extracranial JFS with extension into JF	Extracranial JFS with minor extension to the bone
D	Dumbbell-shaped JFS with both intra- and extracranial parts	Saddle-bag-shaped tumor with intra- and extracranial parts

**Table 2 cancers-13-04218-t002:** Demographics, clinical presentation, and tumor characteristics.

Mdn. Age (Years).		43	(20–71)
Sex	Female	7	(77.8%)
	Male	2	(22.2%)
Preoperative deficits			
Hoarseness		3	(33.3%)
Dysphagia		4	(44.4%)
Cerebellar dysfunction		4	(44.4%)
Extinguished gag reflex		2	(22.2%)
Headache		4	(44.4%)
Hearing loss		6	(66.6%)
Facial nerve palsy		3	(33.3%)
Hypoglossal nerve palsy		2	(22.2%)
Tumor type and approach		Retrosigmoid	ELITE
A	3 (33.3%)	3	
B	1 (0.0%)		1
C	1 (11.1%)		1
D	4 (44.4%)	3	1
Tumor origin			
CN IX		5	(55.6%)
CN X		2	(22.2%)
CN XI		0	0%
Mixed involvement (particularly CN X and XI)		2	(22.2%)
Mdn. Tumor Volume (cm3)		7.08	[3.27–50.1]
Mdn. max. Tumor Diameter (cm)		3.10	[2.4–5.7]
Mean max. Tumor Diameter (cm)		3.45	

Data shown as *n*, = number (%); Mdn., median [range]; CN, cranial nerve; ELITE, extreme lateral infrajugular transcondylar approach.

**Table 3 cancers-13-04218-t003:** Postoperative outcome.

Extent of Resection		
GTR	9	(100%)
Postoperative outcome (permanent)		
Follow-up (months)	16.5	(3–84)
Surgery-related mortality	0	(0.0%)
New hoarseness	1	(11.1%)
New hearing loss	1	(11.1%)
New facial nerve palsy	1	(11.1%)
Shunt-dependent hydrocephalus	2	(22.2%)

Data shown as *n*, number (%); Mdn., median [range]; and GTR, gross total resection.

## Data Availability

The data presented in this study are available on request from the corresponding author.

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
