# Peer review of "Surgical Management of Jugular Foramen Schwannomas"

_cancers, 2021, doi:10.3390/cancers13164218_

Round 1
Reviewer 1 Report
The manuscript “Surgical Management of Jugular Foramen Schwannomas” has carefully been reviewed.
This is a well written article focusing on a rather rare pathological and neurosurgical entity with several surgical problems and risks of post-operative neurological deterioration. However, several considerations must be made:
- Although schwannomas of the jugular foramen are rare, the number of the studied cases is rather small, also considering the overall cases arising from the IX, X and XI cranial nerves. However, the cohort is sufficient to draw interesting considerations.
- In the section “Material and Methods” the surgical approaches are described too extensively. The retrosigmoid approach is well known and does not need to be described. The description of the extreme lateral intrajugular transcondylar approach must be shortened. Besides, the modifications of this approach (lines 101-109) and the citations of the literature references must be eliminated from this section and included in the Discussion.
- In the “Results” section only the clinical symptoms are described. Besides, the number of cases complaining each symptom is very small (2 to 6) to drawn significant percentages. The authors must also include other important data, such as the tumor aspect on MRI (solid, cystic), the size of the schwannomas and the intraoperative data (nerve of origin, arterial relationship etc). The tumor sizes are a determinant factor for surgery of posterior fossa schwannomas, including those of the acoustic, trigeminal and last cranial nerves. How much the size of the tumor influences the surgery?
- The nerve of tumor origin should be defined from the clinical and intraoperative data.
In conclusion, the article may be published after a revision, considering the above discussed modifications.
Reviewer 2 Report
Abstract:
- Please check, if the numbers in perentheses are needed
- Could you please report the clinical outcome of the patient in brevity (only one CN worsening), the rest with slight improvment - that`s excellent
Introduction:
1st page: "Surgerey involving this area usually places the lower... at risk". - Please check for grammar/style, I think it`s "puts"
Reviewer 3 Report
The authors present a retrospective investigation of 9 patients that underwent retrosigmoid or ELITE surgical approaches. There was a gross total resection of all tumors and a median follow-up of 16.5 months. With these approaches they describe a relatively low complications overall. They describe their surgical approach in detail and contrast it to radiation responses in the literature. The authors also describe additional surgical techniques that can be utilized for individual patients. They conclude that surgical resection can be safe with minimal morbidity. Main limitations are unmatched controls, small sample size, short follow-up. The images/figures were well done.
Because jugular foramen schwannomas are less common, the case series will add to the few published studies on this neoplasm. The manuscript lacks a few important details when studying jugular foramen schwannomas:
- What is the origin of the tumor ? (cranial nerve 9, 10, 11?)
- Can complete resection of the tumor be achieved with cranial nerve preservation if the schwannoma significantly involves the nerve? Please discuss
- Hoarseness can improve even if vocal cord paresis or vocal cord paralysis was seen due to compensation of opposite vocal cord. Was a fiberoptic laryngoscopy performed to document vocal cord movement?
- Authors did not discuss "observation" with serial imaging and the advantages of this approach and in what population?
It would be nice to describe how the author's outcomes compares to the large review of jugular foramen schwannomas published by:
Bakar B. The jugular foramen schwannomas: review of the large surgical series. J Korean Neurosurg Soc. 2008;44(5):285-294. doi:10.3340/jkns.2008.44.5.285
Mortality was one outcome of this large series along with stroke, need for tracheotomy, etc. This authors need to better explain the disadvantages of surgery and present a more balanced viewpoint of surgery for these tumors.
If possible, contrasting this cohort to an institutional cohort that was observed or radiated would improve this manuscript significantly.
Round 2
Reviewer 3 Report
The authors present a small case series of jugular foramen schwannomas treated with surgical resection. The authors describe their surgical technique which in their hands produced satisfactory outcomes. The authors state that with these surgical techniques that surgery is a superior treatment.
In the revised submission, the authors updated the manuscript and added some information about the role of observation, radiosurgery, and microsurgical resection mainly to the discussion. With a small case series, the authors declare that surgery is superior to other modalities of treatment (observation and radiosurgery). Scientifically, the study design, results, and discussion (mostly extrapolated from an inadequate review of the vestibular schwannoma literature) do not support that strong claim. The strong statements made in the discussion reads as a surgeon's bias for surgery for large tumors rather than a scientific piece that critically analyzes their contribution to the literature, nuances in treatment decision making for these these tumors overall, and how patient-related factors may influence their decision making and treatment algorithms (e.g. history of COPD or myocardioinfarction). Furthermore, there are grammatical errors in the revised portions of the manuscript that need to be fixed.
It would be in the authors best interest to remove the statements that reflect a "belief" that "surgery is superior". This tone is seen throughout the discussion and even in the last statement of conclusion. The study is simply a case series of 9 jugular foramen schwannomas treated with surgery with a description of surgical techniques. For scientific soundness and to eliminate bias, the language and discussion need to be toned down and modified
Author Response
Thank you very much for your further comments. You are right, as this manuscript is written from a surgical point of view, a certain bias cannot be denied. The tone will be changed, to remain as neutral as possible.
Nevertheless, we hope to fulfil your expectations a bit. Our series is small, but we still do hope to contribute to this subject with our manuscript.
Changes are marked again.
Thank you very much!